# Symplectic Adjoint Method for Exact Gradient of Neural ODE with Minimal Memory

**Takashi Matsubara**
Osaka University
Osaka, Japan 560–8531
matsubara@sys.es.osaka-u.ac.jp

**Yuto Miyatake**
Osaka University
Osaka, Japan 560–0043
miyatake@cas.cmc.osaka-u.ac.jp

**Takaharu Yaguchi**
Kobe University
Kobe, Japan 657–8501
yaguchi@pearl.kobe-u.ac.jp

## Abstract

A neural network model of a differential equation, namely neural ODE, has enabled the learning of continuous-time dynamical systems and probabilistic distributions with high accuracy. The neural ODE uses the same network repeatedly during a numerical integration. The memory consumption of the backpropagation algorithm is proportional to the number of uses *times* the network size. This is true even if a checkpointing scheme divides the computation graph into sub-graphs. Otherwise, the adjoint method obtains a gradient by a numerical integration backward in time. Although this method consumes memory only for a single network use, it requires high computational cost to suppress numerical errors. This study proposes the symplectic adjoint method, which is an adjoint method solved by a symplectic integrator. The symplectic adjoint method obtains the exact gradient (up to rounding error) with memory proportional to the number of uses *plus* the network size. The experimental results demonstrate that the symplectic adjoint method consumes much less memory than the naive backpropagation algorithm and checkpointing schemes, performs faster than the adjoint method, and is more robust to rounding errors.

## 1   Introduction

Deep neural networks offer remarkable methods for various tasks, such as image recognition [18] and natural language processing [4]. These methods employ a residual architecture [21, 34], in which the output $x_{n+1}$ of the $n$-th operation is defined as the sum of a subroutine $f_n$ and the input $x_n$ as $x_{n+1} = f_n(x_n) + x_n$. The residual architecture can be regarded as a numerical integration applied to an ordinary differential equation (ODE) [30]. Accordingly, a neural network model of the differential equation $dx/dt = f(x)$, namely, neural ODE, was proposed in [2]. Given an initial condition $x(0) = x$ as an input, the neural ODE solves an initial value problem by numerical integration, obtaining the final value as an output $y = x(T)$. The neural ODE can model continuous-time dynamics such as irregularly sampled time series [24], stable dynamical systems [37, 42], and physical phenomena associated with geometric structures [3, 13, 31]. Further, because the neural ODE approximates a diffeomorphism [43], it can model probabilistic distributions of real-world data by a change of variables [12, 23, 25, 45].

For an accurate integration, the neural ODE must employ a small step size and a high-order numerical integrator composed of many internal stages. A neural network $f$ is used at each stage of each

35th Conference on Neural Information Processing Systems (NeurIPS 2021).

Table 1: Comparison of the proposed method with existing methods

| Methods | Gradient Calculation | Exact | Checkpoints | Memory Consumption | | Computational Cost |
|---|---|---|---|---|---|---|
| | | | | checkpoint | backprop. | |
| NODE [2] | adjoint method | no | $x_N$ | $M$ | $L$ | $M(N+2\tilde{N})sL$ |
| NODE [2] | backpropagation | yes | — | — | $MNsL$ | $2MNsL$ |
| baseline scheme | backpropagation | yes | $x_0$ | $M$ | $NsL$ | $3MNsL$ |
| ACA [46] | backpropagation | yes | $\{x_n\}_{n=0}^{N-1}$ | $MN$ | $sL$ | $3MNsL$ |
| MALI [47][*] | backpropagation | yes | $x_N$ | $M$ | $sL$ | $4MNsL$ |
| proposed[**] | symplectic adjoint method | yes | $\{x_n\}_{n=0}^{N-1}, \{X_{n,i}\}_{i=1}^{s}$ | $MN+s$ | $L$ | $4MNsL$ |

[*]Available only for the asynchronous leapfrog integrator. [**]Available for any Runge–Kutta methods.

time step. Thus, the backpropagation algorithm consumes exorbitant memory to retain the whole computation graph [39, 2, 10, 46]. The neural ODE employs the adjoint method to reduce memory consumption—this method obtains a gradient by a backward integration along with the state $x$, without consuming memory for retaining the computation graph over time [8, 17, 41, 44]. However, this method incurs high computational costs to suppress numerical errors. Several previous works employed a checkpointing scheme [10, 46, 47]. This scheme only sparsely retains the state $x$ as checkpoints and recalculates a computation graph from each checkpoint to obtain the gradient. However, this scheme still consumes a significant amount of memory to retain the computation graph between checkpoints.

To address the above limitations, this study proposes the *symplectic adjoint method*. The main advantages of the proposed method are presented as follows.

**Exact Gradient and Fast Computation:** In discrete time, the adjoint method suffers from numerical errors or needs a smaller step size. The proposed method uses a specially designed integrator that obtains the exact gradient in discrete time. It works with the same step size as the forward integration and is thus faster than the adjoint method in practice.

**Minimal Memory Consumption:** Excepting the adjoint method, existing methods apply the backpropagation algorithm to the computation graph of the whole or a subset of numerical integration [10, 46, 47]. The memory consumption is proportional to the number of steps/stages in the graph *times* the neural network size. Conversely, the proposed method applies the algorithm only to each use of the neural network, and thus the memory consumption is only proportional to the number of steps/stages *plus* the network size.

**Robust to Rounding Error:** The backpropagation algorithm accumulates the gradient from each use of the neural network and tends to suffer from rounding errors. Conversely, the proposed method obtains the gradient from each step as a numerical integration and is thus more robust to rounding errors.

## 2 Background and Related Work

### 2.1 Neural Ordinary Differential Equation and Adjoint Method

We use the following notation.

$M$: the number of stacked neural ODE components,
$L$: the number of layers in a neural network,
$N, \tilde{N}$: the number of time steps in the forward and backward integrations, respectively, and
$s$: the number of uses of a neural network $f$ per step.

$s$ is typically equal to the number of internal stages of a numerical integrator [17]. A numerical integration forward in time requires a computational cost of $O(MNsL)$. It also provides a computation graph over time steps, which is retained with a memory of $O(MNsL)$; the backpropagation algorithm is then applied to obtain the gradient. The total computational cost is $O(2MNsL)$, where we suppose the computational cost of the backpropagation algorithm is equal to that of forward propagation. The memory consumption and computational cost are summarized in Table 1.

To reduce the memory consumption, the original study on the neural ODE introduced the adjoint method [2, 8, 17, 41, 44]. This method integrates the pair of the system state $x$ and the adjoint variable $\lambda$ backward in time. The adjoint variable $\lambda$ represents the gradient $\frac{\partial \mathcal{L}}{\partial x}$ of some function $\mathcal{L}$, and the backward integration of the adjoint variable $\lambda$ works as the backpropagation (or more generally the reverse-mode automatic differentiation) in continuous time. The memory consumption is $O(M)$ to retain the final values $x(T)$ of $M$ neural ODE components and $O(L)$ to obtain the gradient of a neural network $f$ for integrating the adjoint variable $\lambda$. The computational cost is at least doubled because of the re-integration of the system state $x$ backward in time. The adjoint method suffers from numerical errors [41, 10]. To suppress the numerical errors, the backward integration often requires a smaller step size than the forward integration (i.e., $\tilde{N} > N$), leading to an increase in computation time. Conversely, the proposed symplectic adjoint method uses a specially designed integrator, which provides the exact gradient with the same step size as the forward integration.

## 2.2 Checkpointing Scheme

The checkpointing scheme has been investigated to reduce the memory consumption of neural networks [14, 15], where intermediate states are retained sparsely as checkpoints, and a computation graph is recomputed from each checkpoint. For example, Gruslys *et al.* applied this scheme to recurrent neural networks [15]. When the initial value $x(0)$ of each neural ODE component is retained as a checkpoint, the initial value problem is solved again before applying the backpropagation algorithm to obtain the gradient of the component. Then, the memory consumption is $O(M)$ for checkpoints and $O(NsL)$ for the backpropagation; the memory consumption is $O(M + NsL)$ in total (see the baseline scheme). ANODE scheme retains each step $\{x_n\}_{n=0}^{N-1}$ as a checkpoint with a memory of $O(MN)$ [10]. Form each checkpoint $x_n$, this scheme recalculates the next step $x_{n+1}$ and obtains the gradient using the backpropagation algorithm with a memory of $O(sL)$; the memory consumption is $O(MN + sL)$ in total. ACA scheme improves ANODE scheme for methods with adaptive time-stepping by discarding the computation graph to find an optimal step size. Even with checkpoints, the memory consumption is still proportional to the number of uses $s$ of a neural network $f$ per step, which is not negligible for a high-order integrator, e.g., $s = 6$ for the Dormand–Prince method [7]. In this context, the proposed method is regarded as a checkpointing scheme inside a numerical integrator. Note that previous studies did not use the notation $s$.

Instead of a checkpointing scheme, MALI employs an asynchronous leapfrog (ALF) integrator after the state $x$ is paired up with the velocity state $v$ [47]. The ALF integrator is time-reversible, i.e., the backward integration obtains the state $x$ equal to that in the forward integration without checkpoints [17]. However, the ALF integrator is a second-order integrator, implying that it requires a small step size and a high computational cost to suppress numerical errors. Higher-order Runge–Kutta methods cannot be used in place of the ALF integrator because they are implicit or non-time-reversible. The ALF integrator is inapplicable to physical systems without velocity such as partial differential equation (PDE) systems. Nonetheless, a similar approach named RevNet was proposed before in [11]. When regarding ResNet as a forward Euler method [2, 18], RevNet has an architecture regarded as the leapfrog integrator, and it recalculates the intermediate activations in the reverse direction.

## 3 Adjoint Method

Consider a system

$$\frac{\mathrm{d}}{\mathrm{d}t}x = f(x, t, \theta), \tag{1}$$

where $x$, $t$, and $\theta$, respectively, denote the system state, an independent variable (e.g., time), and parameters of the function $f$. Given an initial condition $x(0) = x_0$, the solution $x(t)$ is given by

$$x(t) = x_0 + \int_0^t f(x(\tau), \tau, \theta)\mathrm{d}\tau. \tag{2}$$

The solution $x(t)$ is evaluated at the terminal $t = T$ by a function $\mathcal{L}$ as $\mathcal{L}(x(T))$. Our main interest is in obtaining the gradients of $\mathcal{L}(x(T))$ with respect to the initial condition $x_0$ and the parameters $\theta$.

Now, we introduce the adjoint method [2, 8, 17, 41, 44]. We first focus on the initial condition $x_0$ and omit the parameters $\theta$. The adjoint method is based on the *variational variable* $\delta(t)$ and the *adjoint*

*variable* $\lambda(t)$. The variational and adjoint variables respectively follow the variational system and adjoint system as follows.

$$\frac{\mathrm{d}}{\mathrm{d}t}\delta(t) = \frac{\partial f}{\partial x}(x(t), t)\delta(t) \text{ for } \delta(0) = I, \quad \frac{\mathrm{d}}{\mathrm{d}t}\lambda(t) = -\frac{\partial f}{\partial x}(x, t)^\top \lambda(t) \text{ for } \lambda(T) = \lambda_T. \quad (3)$$

The variational variable $\delta(t)$ represents the Jacobian $\frac{\partial x(t)}{\partial x_0}$ of the state $x(t)$ with respect to the initial condition $x_0$; the detailed derivation is summarized in Appendix A.

**Remark 1.** *The quantity $\lambda^\top \delta$ is time-invariant, i.e., $\lambda(t)^\top \delta(t) = \lambda(0)^\top \delta(0)$.*

The proofs of most Remarks and Theorems in this paper are summarized in Appendix B.

**Remark 2.** *The adjoint variable $\lambda(t)$ represents the gradient $(\frac{\partial \mathcal{L}(x(T))}{\partial x(t)})^\top$ if the final condition $\lambda_T$ of the adjoint variable $\lambda$ is set to $(\frac{\partial \mathcal{L}(x(T))}{\partial x(T)})^\top$.*

This is because of the chain rule. Thus, the backward integration of the adjoint variable $\lambda(t)$ works as reverse-mode automatic differentiation. The adjoint method has been used for data assimilation, where the initial condition $x_0$ is optimized by a gradient-based method. For system identification (i.e., parameter adjustment), one can consider the parameters $\theta$ as a part of the augmented state $\tilde{x} = [x \ \theta]^\top$ of the system

$$\frac{\mathrm{d}}{\mathrm{d}t}\tilde{x} = \tilde{f}(\tilde{x}, t), \ \tilde{f}(\tilde{x}, t) = \begin{bmatrix} f(x, t, \theta) \\ 0 \end{bmatrix}, \ \tilde{x}(0) = \begin{bmatrix} x_0 \\ \theta \end{bmatrix}. \quad (4)$$

The variational and adjoint variables are augmented in the same way. Hereafter, we let $x$ denote the state or augmented state without loss of generality. See Appendix C for details.

According to the original implementation of the neural ODE [2], the final value $x(T)$ of the system state $x$ is retained after forward integration, and the pair of the system state $x$ and the adjoint variable $\lambda$ is integrated backward in time to obtain the gradients. The right-hand sides of the main system in Eq. (1) and the adjoint system in Eq. (3) are obtained by the forward and backward propagations of the neural network $f$, respectively. Therefore, the computational cost of the adjoint method is twice that of the ordinary backpropagation algorithm.

After a numerical integrator discretizes the time, Remark 1 does not hold, and thus the adjoint variable $\lambda(t)$ is not equal to the exact gradient [10, 41]. Moreover, in general, the numerical integration backward in time is not consistent with that forward in time. Although a small step size (i.e., a small tolerance) suppresses numerical errors, it also leads to a longer computation time. These facts provide the motivation to obtain the exact gradient with a small memory, in the present study.

## 4 Symplectic Adjoint Method

### 4.1 Runge–Kutta Method

We first discretize the main system in Eq. (1). Let $t_n$, $h_n$, and $x_n$ denote the $n$-th time step, step size, and state, respectively, where $h_n = t_{n+1} - t_n$. Previous studies employed one of the Runge–Kutta methods, generally expressed as

$$x_{n+1} = x_n + h_n \sum_{i=1}^{s} b_i k_{n,i},$$
$$k_{n,i} := f(X_{n,i}, t_n + c_i h_n), \quad (5)$$
$$X_{n,i} := x_n + h_n \sum_{j=1}^{s} a_{i,j} k_{n,j}.$$

The coefficients $a_{i,j}$, $b_i$, and $c_i$ are summarized as the Butcher tableau [16, 17, 41]. If $a_{i,j} = 0$ for $j \geq i$, the intermediate state $X_{n,i}$ is calculable from $i = 1$ to $i = s$ sequentially; then, the Runge–Kutta method is considered explicit. Runge–Kutta methods are not time-reversible in general, i.e., the numerical integration backward in time is not consistent with that forward in time.

**Remark 3** (Bochev and Scovel [1], Hairer et al. [16]). *When the system in Eq. (1) is discretized by the Runge–Kutta method in Eq. (5), the variational system in Eq. (3) is discretized by the same Runge–Kutta method.*

Therefore, it is not necessary to solve the variational variable $\delta(t)$ separately.

## 4.2 Symplectic Runge–Kutta Method for Adjoint System

We assume $b_i \neq 0$ for $i = 1, \ldots, s$. We suppose the adjoint system to be solved by another Runge–Kutta method with the same step size as that used for the system state $x$, expressed as

$$\lambda_{n+1} = \lambda_n + h_n \sum_{i=1}^{s} B_i l_{n,i},$$

$$l_{n,i} := -\frac{\partial f}{\partial x}(X_{n,i}, t_n + C_i h_n)^\top \Lambda_{n,i}, \tag{6}$$

$$\Lambda_{n,i} := \lambda_n + h_n \sum_{j=1}^{s} A_{i,j} l_{n,j}.$$

The final condition $\lambda_N$ is set to $\left(\frac{\partial \mathcal{L}(x_N)}{\partial x_N}\right)^\top$. Because the time evolutions of the variational variable $\delta$ and the adjoint variable $\lambda$ are expressible by two equations, the combined system is considered as a partitioned system. A combination of two Runge–Kutta methods for solving a partitioned system is called a partitioned Runge–Kutta method, where $C_i = c_i$ for $i = 1, \ldots, s$. We introduce the following condition for a partitioned Runge–Kutta method.

**Condition 1.** $b_i A_{i,j} + B_j a_{j,i} - b_i B_j = 0$ *for* $i, j = 1, \ldots, s$, *and* $B_i = b_i \neq 0$ *and* $C_i = c_i$ *for* $i = 1, \ldots, s$.

**Theorem 1** (Sanz-Serna [41])**.** *The partitioned Runge–Kutta method in Eqs.* (5) *and* (6) *conserves a bilinear quantity* $S(\delta, \lambda)$ *if the continuous-time system conserves the quantity* $S(\delta, \lambda)$ *and Condition 1 holds.*

Because the bilinear quantity $S$ (including $\lambda^\top \delta$) is conserved, the adjoint system solved by the Runge–Kutta method in Eq. (6) under Condition 1 provides the exact gradient as the adjoint variable $\lambda_n = \left(\frac{\partial \mathcal{L}(x_N)}{\partial x_n}\right)^\top$. The Dormand–Prince method, one of the most popular Runge–Kutta methods, has $b_2 = 0$ [7]. For such methods, the Runge–Kutta method under Condition 1 in Eq. (6) is generalized as

$$\lambda_{n+1} = \lambda_n + h_n \sum_{i=1}^{s} \tilde{b}_i l_{n,i},$$

$$l_{n,i} := -\frac{\partial f}{\partial x}(X_{n,i}, t_n + c_i h_n)^\top \Lambda_{n,i}, \tag{7}$$

$$\Lambda_{n,i} := \begin{cases} \lambda_n + h_n \sum_{j=1}^{s} \tilde{b}_j \left(1 - \frac{a_{j,i}}{b_i}\right) l_{n,j} & \text{if } i \notin I_0 \\ -\sum_{j=1}^{s} \tilde{b}_j a_{j,i} l_{n,j} & \text{if } i \in I_0, \end{cases}$$

where

$$\tilde{b}_i = \begin{cases} b_i & \text{if } i \notin I_0 \\ h_n & \text{if } i \in I_0, \end{cases} \quad I_0 = \{i | i = 1, \ldots, s, \ b_i = 0\}. \tag{8}$$

Note that this numerical integrator is no longer a Runge–Kutta method and is an alternative expression for the "fancy" integrator proposed in [41].

**Theorem 2.** *The combination of the integrators in Eqs.* (5) *and* (7) *conserves a bilinear quantity* $S(\delta, \lambda)$ *if the continuous-time system conserves the quantity* $S(\delta, \lambda)$.

**Remark 4.** *The Runge–Kutta method in Eq.* (6) *under Condition 1 and the numerical integrator in Eq.* (7) *are explicit backward in time if the Runge–Kutta method in Eq.* (5) *is explicit forward in time.*

We emphasize that Theorems 1 and 2 hold for any ODE systems even if the systems have discontinuity [19], stochasticity [29], or physics constraints [13]. This is because the Theorems are not the properties of a system but of Runge–Kutta methods.

A partitioned Runge–Kutta method that satisfies Condition 1 is symplectic [17, 16]. It is known that, when a symplectic integrator is applied to a Hamiltonian system using a fixed step size, it conserves a modified Hamiltonian, which is an approximation to the system energy of the Hamiltonian system. The bilinear quantity $S$ is associated with the symplectic structure but not with a Hamiltonian. Regardless of the step size, a symplectic integrator conserves the symplectic structure and thereby conserves the bilinear quantity $S$. Hence, we named this method the *symplectic adjoint method*. For integrators other than Runge–Kutta methods, one can design the integrator for the adjoint system so that the pair of integrators is symplectic (see [32] for example).

---
**Algorithm 1** Forward Integration
---
**Input:** $x_0$
**Output:** $x_N$ ,$\{x_n\}_{n=0}^{N-1}$
1: **for** $n = 0$ to $N - 1$ **do**
2:    Retain $x_n$ as a checkpoint
    *According to Eq.* (5)
3:    **for** $i = 1$ to $s$ **do**
4:       Get $X_{n,i}$ using $x_n$ and $k_{n,j}$ for $j < i$
5:       Get $k_{n,i}$ using $X_{n,i}$
6:    **end for**
7:    Get $x_{n+1}$ using $x_n$ and $k_{n,i}$
8: **end for**
---

---
**Algorithm 2** Backward Integration
---
**Input:** $x_N$ ,$\{x_n\}_{n=0}^{N-1}$
**Output:** $\lambda_0$
1: **for** $n = N - 1$ to $0$ **do**
2:    Load checkpoint $x_n$
    *According to Eq.* (5)
3:    **for** $i = 1$ to $s$ **do**
4:       Get $X_{n,i}$ using $x_n$ and $k_{n,j}$ for $j < i$
5:       Get $k_{n,i}$ using $X_{n,i}$.
6:       Retain $X_{n,i}$ as a checkpoint
7:    **end for**
    *According to Eq.* (7)
8:    **for** $i = s$ to $1$ **do**
9:       Get $\Lambda_{n,i}$ using $\lambda_{n+1}$ and $l_{n,j}$ for $j > i$
10:      Load checkpoint $X_{n,i}$
11:      Get $l_{n,i}$ using $\Lambda_{n,i}$ and $X_{n,i}$.
12:      Discard checkpoint $X_{n,i}$
13:    **end for**
14:    Get $\lambda_n$ using $\lambda_{n+1}$ and $l_{n,i}$
15:    Discard checkpoint $x_n$
16: **end for**
---

### 4.3 Proposed Implementation

The theories given in the last section were mainly introduced for the numerical analysis in [41]. Because the original expression includes recalculations of intermediate variables, we propose the alternative expression in Eq. (7) to reduce the computational cost. The discretized adjoint system in Eq. (7) depends on the vector–Jacobian product (VJP) $\Lambda^\top \frac{\partial f}{\partial x}$. To obtain it, the computation graph from the input $X_{n,i}$ to the output $f(X_{n,i}, t_n + c_i h_n)$ is required. When the computation graph in the forward integration is entirely retained, the memory consumption and computational cost are of the same orders as those for the naive backpropagation algorithm. To reduce the memory consumption, we propose the following strategy as summarized in Algorithms 1 and 2.

At the forward integration of a neural ODE component, the pairs of system states $x_n$ and time points $t_n$ at time steps $n = 0, \ldots, N - 1$ are retained with a memory of $O(N)$ as checkpoints, and all computation graphs are discarded, as shown in Algorithm 1. For $M$ neural ODE components, the memory for checkpoints is $O(MN)$. The backward integration is summarized in Algorithm 2. The below steps are repeated from $n = N - 1$ to $n = 0$. From the checkpoint $x_n$, the intermediate states $X_{n,i}$ for $s$ stages are obtained following the Runge–Kutta method in Eq. (5) and retained as checkpoints with a memory of $O(s)$, while all computation graphs are discarded. Then, the adjoint system is integrated from $n + 1$ to $n$ using Eq. (7). Because the computation graph of the neural network $f$ in line 5 is discarded, it is recalculated and the VJP $\lambda^\top \frac{\partial f}{\partial x}$ is obtained using the backpropagation algorithm one-by-one in line 11, where only a single use of the neural network is recalculated at a time. This is why the memory consumption is proportional to the number of checkpoints $MN + s$ *plus* the neural network size $L$. By contrast, existing methods apply the backpropagation algorithm to the computation graph of a single step composed of $s$ stages or multiple steps. The memory consumption is proportional to the number of uses of the neural network between two checkpoints ($s$ at least) *times* the neural network size $L$, in addition to the memory for checkpoints (see Table 1). Due to the recalculation, the computational cost of the proposed strategy is $O(4MNsL)$, whereas those of the adjoint method [2] and ACA [46] are $O(M(N + 2\tilde{N})sL)$ and $O(3MNsL)$, respectively. However, the increase in the computation time is much less than that expected theoretically because of other bottlenecks (as demonstrated later).

## 5 Experiments

We evaluated the performance of the proposed symplectic adjoint method and existing methods using PyTorch 1.7.1 [35]. We implemented the proposed symplectic adjoint method by extending the adjoint method implemented in the package torchdiffeq 0.1.1 [2]. We re-implemented ACA [46] because the interfaces of the official implementation is incompatible with torchdiffeq. In practice, the number of checkpoints for an integration can be varied; we implemented a baseline scheme that retains only a single checkpoint per neural ODE component. The source code is available at https://github.com/tksmatsubara/symplectic-adjoint-method.

Table 2: Results obtained for continuous normalizing flows.

| | MINIBOONE ($M = 1$) | | | GAS ($M = 5$) | | | POWER ($M = 5$) | | |
|---|---|---|---|---|---|---|---|---|---|
| | NLL | mem. | time | NLL | mem. | time | NLL | mem. | time |
| adjoint method [2] | 10.59±0.17 | 170 | 0.74 | -10.53±0.25 | 24 | 4.82 | -0.31±0.01 | **8.1** | 6.33 |
| backpropagation [2] | 10.54±0.18 | 4436 | 0.91 | -9.53±0.42 | 4479 | 12.00 | -0.24±0.05 | 1710.9 | 10.64 |
| baseline scheme | 10.54±0.18 | 4457 | 1.10 | -9.53±0.42 | 1858 | 5.48 | -0.24±0.05 | 515.2 | 4.37 |
| ACA [46] | 10.57±0.30 | 306 | 0.77 | -10.65±0.45 | 73 | 3.98 | -0.31±0.02 | 29.5 | 5.08 |
| proposed | 10.49±0.11 | **95** | 0.84 | -10.89±0.11 | **20** | 4.39 | -0.31±0.02 | 9.2 | 5.73 |

| | HEPMASS ($M = 10$) | | | BSDS300 ($M = 2$) | | | MNIST ($M = 6$) | | |
|---|---|---|---|---|---|---|---|---|---|
| | NLL | mem. | time | NLL | mem. | time | NLL | mem. | time |
| adjoint method [2] | 16.49±0.25 | 40 | 4.19 | -152.04±0.09 | 577 | 11.70 | 0.918±0.011 | 1086 | 10.12 |
| backpropagation [2] | 17.03±0.22 | 5254 | 11.82 | — | — | — | — | — | — |
| baseline scheme | 17.03±0.22 | 1102 | 4.40 | — | — | — | — | — | — |
| ACA [46] | 16.41±0.39 | 88 | 3.67 | -151.27±0.47 | 757 | 6.97 | 0.919±0.003 | 4332 | 7.94 |
| proposed | 16.48±0.20 | **35** | 4.15 | -151.17±0.15 | **283** | 8.07 | 0.917±0.002 | **1079** | 9.42 |

Negative log-likelihoods (NLL), peak memory consumption [MiB], and computation time per iteration [s/itr]. See Table A2 in Appendix for standard deviations.

## 5.1 Continuous Normalizing Flow

**Experimental Settings:** We evaluated the proposed symplectic adjoint method on training continuous normalizing flows [12]. A normalizing flow is a neural network that approximates a bijective map $g$ and obtains the exact likelihood of a sample $u$ by the change of variables $\log p(u) = \log p(z) + \log |\det \frac{\partial g(u)}{\partial u}|$, where $z = g(u)$ and $p(z)$ denote the corresponding latent variable and its prior, respectively [5, 6, 38]. A continuous normalizing flow is a normalizing flow whose map $g$ is modeled by stacked neural ODE components, in particular, $u = x(0)$ and $z = x(T)$ for the case with $M = 1$. The log-determinant of the Jacobian is obtained by a numerical integration together with the system state $x$ as $\log |\det \frac{\partial g(u)}{\partial u}| = -\int_0^T \text{Tr}(\frac{\partial f}{\partial x}(x(t), t)) \mathrm{d}t$. The trace operation Tr is approximated by the Hutchinson estimator [22]. We adopted the experimental settings of the continuous normalizing flow, FFJORD[1] [12], unless stated otherwise.

We examined five real tabular datasets, namely, MiniBooNE, GAS, POWER, HEPMASS, and BSDS300 datasets [33]. The network architectures were the same as those that achieved the best results in the original experiments; the number of neural ODE components $M$ varied across datasets. We employed the Dormand–Prince integrator, which is a fifth-order Runge–Kutta method with adaptive time-stepping, composed of seven stages [7]. Note that the number of function evaluations per step is $s = 6$ because the last stage is reused as the first stage of the next step. We set the absolute and relative tolerances to atol $= 10^{-8}$ and rtol $= 10^{-6}$, respectively. The neural networks were trained using the Adam optimizer [27] with a learning rate of $10^{-3}$. We used a batch-size of 1000 for all datasets to put a mini-batch into a single NVIDIA GeForce RTX 2080Ti GPU with 11 GB of memory, while the original experiments employed a batch-size of 10 000 for the latter three datasets on multiple GPUs. When using multiple GPUs, bottlenecks such as data transfer across GPUs may affect performance, and a fair comparison becomes difficult. Nonetheless, the naive backpropagation algorithm and baseline scheme consumed the entire memory for BSDS300 dataset.

We also examined the MNIST dataset [28] using a single NVIDIA RTX A6000 GPU with 48 GB of memory. Following the original study, we employed the multi-scale architecture and set the tolerance to atol $=$ rtol $= 10^{-5}$. We set the learning rate to $10^{-3}$ and then reduced it to $10^{-4}$ at the 250th epoch. While the original experiments used a batch-size of 900, we set the batch-size to 200 following the official code[1]. The naive backpropagation algorithm and baseline scheme consumed the entire memory.

**Performance:** The medians $\pm$ standard deviations of three runs are summarized in Table 2. In many cases, all methods achieved negative log-likelihoods (NLLs) with no significant difference

---

[1] https://github.com/rtqichen/ffjord (MIT License)

because all but the adjoint method provide the exact gradients up to rounding error, and the adjoint method with a small tolerance provides a sufficiently accurate gradient. The naive backpropagation algorithm and baseline scheme obtained slightly worse results on the GAS, POWER, and HEPMASS datasets. Due to adaptive time-stepping, the numerical integrator sometimes makes the step size much smaller, and the backpropagation algorithm over time steps suffered from rounding errors. Conversely, ACA and the proposed symplectic adjoint method applied the backpropagation algorithm separately to a subset of the integration, thereby becoming more robust to rounding errors (see Appendix D.1 for details).

After the training procedure, we obtained the peak memory consumption during additional training iterations (mem. [MiB]), from which we subtracted the memory consumption before training (i.e., occupied by the model parameters, loaded data, etc.). The memory consumption still includes the optimizer's states and the intermediate results of the multiply–accumulate operation. The results roughly agree with the theoretical orders shown in Table 1 (see also Table A2 for standard deviations). The symplectic adjoint method consumed much smaller memory than the naive backpropagation algorithm and the checkpointing schemes. Owing to the optimized implementation, the symplectic adjoint method consumed smaller memory than the adjoint method in some cases (see Appendix D.2).

On the other hand, the computation time per iteration (time [s/itr]) during the additional training iterations does not agree with the theoretical orders. First, the adjoint method was slower in many cases, especially for the BSDS300 and MNIST datasets. For obtaining the gradients, the adjoint method integrates the adjoint variable $\lambda$, whose size is equal to the sum of the sizes of the parameters $\theta$ and the system state $x$. With more parameters, the probability that at least one parameter does not satisfy the tolerance value is increased. An accurate backward integration requires a much smaller step size than the forward integration (i.e., $\tilde{N}$ much greater than $N$), leading to a longer computation time. Second, the naive backpropagation algorithm and baseline scheme were slower than that expected theoretically, in many cases. A method with high memory consumption may have to wait for a retained computation graph to be loaded or memory to be freed, leading to an additional bottleneck. The symplectic adjoint method is free from the above bottlenecks and performs faster in practice; it was faster than the adjoint method for all but MiniBooNE dataset.

The symplectic adjoint method is superior (or at least competitive) to the adjoint method, naive backpropagation, and baseline scheme in terms of both memory consumption and computation time. Between the proposed symplectic adjoint method and ACA, a trade-off exists between memory consumption and computation time.

**Robustness to Tolerance:** The adjoint method provides gradients with numerical errors. To evaluate the robustness against tolerance, we employed MiniBooNE dataset and varied the absolute tolerance atol while maintaining the relative tolerance as $\mathsf{rtol} = 10^2 \times \mathsf{atol}$. During the training, we obtained the computation time per iteration, as summarized in the upper panel of Fig. 1. The computation time reduced as the tolerance increased. After training, we obtained the NLLs with $\mathsf{atol} = 10^{-8}$, as summarized in the bottom panel of Fig. 1. The adjoint method performed well only with $\mathsf{atol} < 10^{-4}$. With $\mathsf{atol} = 10^{-4}$, the numerical error in the backward integration was non-negligible, and the performance degraded. With $\mathsf{atol} > 10^{-4}$, the adjoint method destabilized. The symplectic adjoint method performed well even with $\mathsf{atol} = 10^{-4}$. Even with $10^{-4} < \mathsf{atol} < 10^{-2}$, it

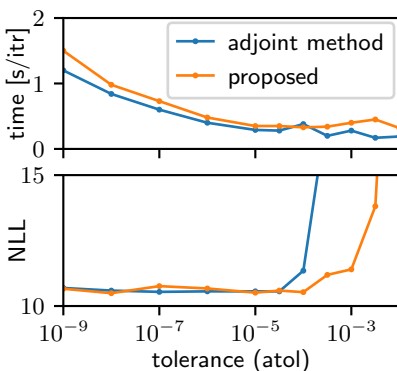

Figure 1: With different tolerances.

performed to a certain level, while the numerical error in the forward integration was non-negligible. Because of the exact gradient, the symplectic adjoint method is robust to a large tolerance compared with the adjoint method, and thus potentially works much faster with an appropriate tolerance.

**Different Runge–Kutta Methods:** The Runge–Kutta family includes various integrators characterized by the Butcher tableau [16, 17, 41], such as the Heun–Euler method (a.k.a. adaptive Heun), Bogacki–Shampine method (a.k.a. bosh3), fifth-order Dormand–Prince method (a.k.a. dopri5), and eighth-order Dormand–Prince method (a.k.a. dopri8). These methods have the orders of $p = 2, 3, 5,$ and 8 using $s = 2, 3, 6,$ and 12 function evaluations, respectively. We examined these methods using

Table 3: Results obtained for GAS dataset with different Runge–Kutta methods.

| | $p=2, s=2$ | | $p=3, s=3$ | | $p=5, s=6$ | | $p=8, s=12$ | |
| --- | --- | --- | --- | --- | --- | --- | --- | --- |
| | mem. | time | mem. | time | mem. | time | mem. | time |
| adjoint method [2] | **21**± 0 | 247.47± 7.52 | **22**±0 | 18.32±0.88 | 24± 0 | 5.34±0.31 | 28± 0 | 9.77±0.81 |
| backpropagation [2] | — | — | — | — | 4433±255 | 11.85±1.10 | — | — |
| baseline scheme | — | — | — | — | 1858±228 | 5.82±0.28 | 4108±576 | 22.76±3.70 |
| ACA [46] | 607±30 | 232.90±13.81 | 69±2 | 17.72±1.38 | 73± 0 | 4.15±0.21 | 138± 0 | 9.36±0.55 |
| proposed | 589±14 | 262.99± 5.19 | 43±2 | 18.59±0.75 | **20**± 0 | 4.78±0.32 | **21**± 0 | 11.41±0.23 |

Peak memory consumption [MiB], and computation time per iteration [s/itr].

GAS dataset, and the results are summarized in Table 3. The naive backpropagation algorithm and baseline scheme consumed the entire memory in some cases, as denoted by dashes. We omit the NLLs because all methods used the same tolerance and achieved the same NLLs.

Compared to ACA, the symplectic adjoint method suppresses the memory consumption more significantly with a higher-order method (i.e., more function evaluations $s$), as the theory suggests in Table 1. With the Heun–Euler method, all methods were extremely slow, and all but the adjoint method consumed larger memory. A lower-order method has to use an extremely small step size to satisfy the tolerance, thereby increasing the number of steps $N$, computation time, and memory for checkpoints. This result indicates the limitations of methods that depend on lower-order integrators, such as MALI [47]. With the eighth-order Dormand–Prince method, the adjoint method performs relatively faster. This is because the backward integration easily satisfies the tolerance with a higher-order method (i.e., $\tilde{N} \simeq N$). Nonetheless, in terms of computation time, the fifth-order Dormand–Prince method is the best choice, for which the symplectic adjoint method greatly reduces the memory consumption and performs faster than all but ACA.

**Memory for Checkpoints:** To evaluate the memory consumption with varying numbers of checkpoints, we used the fifth-order Dormand–Prince method and varied the number of steps $N$ for MNIST by manually varying the step size. We summarized the results in Fig. 2 on a log-log scale. Note that, with the adaptive stepping, FFJORD needs approximately $MN = 200$ steps for MNIST and fewer steps for other datasets. Because we set $\tilde{N} = N$, but $\tilde{N} > N$ in practice, the adjoint method is expected to require a longer computation time.

The memory consumption roughly follows the theoretical orders summarized in Table 1. The adjoint method needs a memory of $O(L)$ for the backpropagation, and the symplectic adjoint method needs an additional memory of $O(MN+s)$ for checkpoints. Until the number of steps $MN$ exceeds a thousand, the memory for checkpoints is negligible compared to that for the backpropagation. Compared to the symplectic adjoint method, ACA needs a memory of $O(sL)$ for the backpropagation

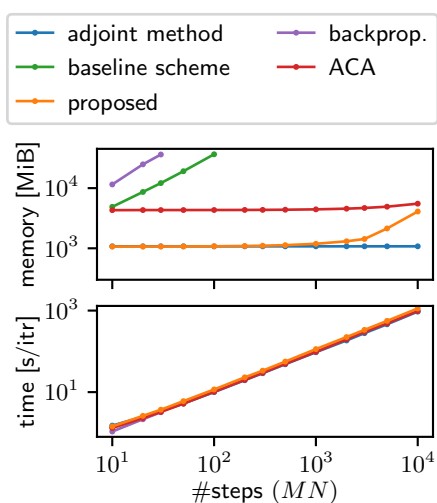

Figure 2: Different number of steps.

over $s$ stages. The increase in memory is significant until the number of steps $MN$ reaches ten thousand. For a stiff (non-smooth) ODE for which a numerical integrator needs thousands of steps, one can employ a higher-order integrator such as the eighth-order Dormand–Prince method and suppress the number of steps. For a stiffer ODE, implicit integrators are commonly used, which are out of the scope of this study and the related works in Table 1. Therefore, we conclude that the symplectic adjoint method needs the memory at the same level as the adjoint method and much smaller than others in practical ranges.

A possible alternative to the proposed implementation in Algorithms 1 and 2 retains all intermediate states $X_{n,i}$ during the forward integration. Its computational cost and memory consumption are $O(3MNsL)$ and $O(MNs + L)$, respectively. The memory for checkpoints can be non-negligible with a practical number of steps.

Table 4: Results obtained for continuous-time physical systems.

| | KdV Equation | | | Cahn–Hilliard System | | |
|---|---|---|---|---|---|---|
| | MSE $(\times 10^{-3})$ | mem. | time | MSE $(\times 10^{-6})$ | mem. | time |
| adjoint method [2] | $1.61\pm3.23$ | $93.7\pm0.0$ | $276\pm16$ | $5.58\pm1.67$ | $93.7\pm0.0$ | $942\pm24$ |
| backpropagation [2] | $1.61\pm3.40$ | $693.9\pm0.0$ | $105\pm 4$ | $4.68\pm1.89$ | $3047.1\pm0.0$ | $425\pm13$ |
| ACA [46] | $1.61\pm3.40$ | $647.8\pm0.0$ | $137\pm 5$ | $5.82\pm2.33$ | $648.0\pm0.0$ | $484\pm13$ |
| proposed | $1.61\pm4.00$ | $\mathbf{79.8}\pm0.0$ | $162\pm 6$ | $5.47\pm1.46$ | $\mathbf{80.3}\pm0.0$ | $568\pm22$ |

Mean-squared errors (MSEs) in long-term predictions, peak memory consumption [MiB], and computation time per iteration [ms/itr].

## 5.2 Continuous-Time Dynamical System

**Experimental Settings:** We evaluated the symplectic adjoint method on learning continuous-time dynamical systems [13, 31, 40]. Many physical phenomena can be modeled using the gradient of system energy $H$ as $\mathrm{d}x/\mathrm{d}t = G\nabla H(x)$, where $G$ is a coefficient matrix that determines the behaviors of the energy [9]. We followed the experimental settings of HNN++, provided in [31][2]. A neural network composed of one convolution layer and two fully connected layers approximated the energy function $H$ and learned the time series by interpolating two successive samples. The deterministic convolution algorithm was enabled (see Appendix D.3 for discussion). We employed two physical systems described by PDEs, namely the Korteweg–De Vries (KdV) equation and the Cahn–Hilliard system. We used a batch-size of 100 to put a mini-batch into a single NVIDIA TITAN V GPU instead of the original batch-size of 200. Moreover, we used the eighth-order Dormand–Prince method [17], composed of 13 stages, to emphasize the efficiency of the proposed method. We omitted the baseline scheme because of $M = 1$. We evaluated the performance using mean squared errors (MSEs) in the system energy for long-term predictions.

**Performance:** The medians $\pm$ standard deviations of 15 runs are summarized in Table 4. Due to the accumulated error in the numerical integration, the MSEs had large variances, but all methods obtained similar MSEs. ACA consumed much more memory than the symplectic adjoint method because of the large number of stages; the symplectic adjoint method is more beneficial for physical simulations, which often require extremely higher-order methods. Due to the severe nonlinearity, the adjoint method had to employ a small step size and thus performed slower than others (i.e., $\tilde{N} > N$).

## 6 Conclusion

We proposed the symplectic adjoint method, which solves the adjoint system by a symplectic integrator with appropriate checkpoints and thereby provides the exact gradient. It only applies the backpropagation algorithm to each use of the neural network, and thus consumes memory much less than the backpropagation algorithm and the checkpointing schemes. Its memory consumption is competitive to that of the adjoint method because the memory consumed by checkpoints is negligible in most cases. The symplectic adjoint method provides the exact gradient with the same step size as that used for the forward integration. Therefore, in practice, it performs faster than the adjoint method, which requires a small step size to suppress numerical errors.

As shown in the experiments, the best integrator and checkpointing scheme may depend on the target system and computational resources. For example, Kim et al. [26] has demonstrated that quadrature methods can reduce the computation cost of the adjoint system for a stiff equation in exchange for the additional memory consumption. Practical packages provide many integrators and can choose the best ones [20, 36]. In the future, we will provide the proposed symplectic adjoint method as a part of such packages for appropriate systems.

## Acknowledgments and Disclosure of Funding

This study was partially supported by JST CREST (JPMJCR1914), JST PRESTO (JPMJPR21C7), and JSPS KAKENHI (19K20344).

---

[2] `https://github.com/tksmatsubara/discrete-autograd` (MIT License)

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
