# Supplementary Material: Appendices

## A   Derivation of Variational System

Let us consider a perturbed initial condition $\bar{x}_0 = x_0 + \bar{\delta}_0$, from which the solution $\bar{x}(t)$ arises. Suppose that the solution $\bar{x}(t)$ satisfies $\bar{x}(t) = x(t) + \bar{\delta}(t)$. Then,

$$\begin{aligned}
\frac{\mathrm{d}}{\mathrm{d}t}\bar{\delta} &= \frac{\mathrm{d}}{\mathrm{d}t}(\bar{x} - x) \\
&= f(\bar{x}, t) - f(x, t) \\
&= \frac{\partial f}{\partial x}(x, t)(\bar{x} - x) + o(|\bar{x} - x|) \\
&= \frac{\partial f}{\partial x}(x, t)\bar{\delta} + o(|\bar{\delta}|),
\end{aligned} \tag{9}$$
$$\bar{\delta}(0) = \bar{\delta}_0.$$

Dividing $\bar{\delta}$ by $\bar{\delta}_0$ and taking the limit as $|\bar{\delta}_0| \to +0$, we define the variational variable as $\delta(t) = \frac{\partial x(t)}{\partial x_0}$ and the *variational system* as

$$\frac{\mathrm{d}}{\mathrm{d}t}\delta(t) = \frac{\partial f}{\partial x}(x(t), t)\delta(t) \text{ for } \delta(0) = I. \tag{10}$$

## B   Complete Proofs

**Proof of Remark 1:**

$$\frac{\mathrm{d}}{\mathrm{d}t}\left(\lambda^\top \delta\right) = \left(\frac{\mathrm{d}}{\mathrm{d}t}\lambda\right)^\top \delta + \lambda^\top \left(\frac{\mathrm{d}}{\mathrm{d}t}\delta\right) = \left(-\frac{\partial f}{\partial x}(x, t)^\top \lambda\right)^\top \delta + \lambda^\top \left(\frac{\partial f}{\partial x}(x, t)\delta\right) = 0. \tag{11}$$

**Proof of Remark 2:**   Because $\delta(t) = \frac{\partial x(t)}{\partial x_0}$ and $\lambda^\top \delta$ is time-invariant,

$$\frac{\partial \mathcal{L}(x(T))}{\partial x_0} = \frac{\partial \mathcal{L}(x(T))}{\partial x(T)}\frac{\partial x(T)}{\partial x_0} = \lambda(T)^\top \delta(T) = \lambda(t)^\top \delta(t) = \frac{\partial \mathcal{L}(x(T))}{\partial x(t)}\frac{\partial x(t)}{\partial x_0}. \tag{12}$$

**Proof of Remark 3:**   Differentiating each term in the Runge–Kutta method in Eq. (5) by the initial condition $x_0$ gives the Runge–Kutta method applied to the variational variable $\delta$, as follows.

$$\begin{aligned}
\delta_{n+1} &= \delta_n + h_n \sum_{i=1}^s b_i d_{n,i}, \\
d_{n,i} &:= \frac{\partial k_{n,i}}{\partial x_0} = \frac{\partial f(X_{n,i}, t_n + c_i h_n)}{\partial x_0} = \frac{\partial f(X_{n,i}, t_n + c_i h_n)}{\partial X_{n,i}}\Delta_{n,i}, \\
\Delta_{n,i} &:= \frac{\partial X_{n,i}}{\partial x_0} = \delta_n + h_n \sum_{j=1}^s a_{i,j} d_{n,j}.
\end{aligned} \tag{13}$$

**Proof of Theorem 1:**   Because the quantity $S$ is conserved in continuous time,

$$\frac{\mathrm{d}}{\mathrm{d}t}S(\delta, \lambda) = 0. \tag{14}$$

Because the quantity $S$ is bilinear,

$$\frac{\mathrm{d}}{\mathrm{d}t}S(\delta, \lambda) = \frac{\partial S}{\partial \delta}\frac{\mathrm{d}\delta}{\mathrm{d}t} + \frac{\partial S}{\partial \lambda}\frac{\mathrm{d}\lambda}{\mathrm{d}t} = S\left(\frac{\mathrm{d}\delta}{\mathrm{d}t}, \lambda\right) + S\left(\delta, \frac{\mathrm{d}\lambda}{\mathrm{d}t}\right), \tag{15}$$

which implies

$$S(d_{n,i}, \Lambda_{n,i}) + S(\Delta_{n,i}, l_{n,i}) = 0. \tag{16}$$

The change in the bilinear quantity $S(\delta, \lambda)$ is

$$
\begin{aligned}
S(\delta_{n+1}, \lambda_{n+1}) - S(\delta_n, \lambda_n) &= S(\delta_n + h_n \sum_i b_i d_{n,i}, \lambda_n + h_n \sum_i B_i l_{n,i}) - S(\delta_n, \lambda_n) \\
&= \sum_i b_i h_n S(d_{n,i}, \lambda_n) + \sum_i B_i h_n S(\delta_n, l_{n,i}) \\
&\quad + \sum_i \sum_j b_i B_j h_n^2 S(d_{n,i}, l_{n,j}) \\
&= \sum_i b_i h_n S(d_{n,i}, \Lambda_{n,i} - h_n \sum_j A_{i,j} l_{n,j}) \\
&\quad + \sum_i B_i h_n S(\Delta_{n,i} - h_n \sum_j a_{i,j} d_{n,j}, l_{n,i}) \\
&\quad + \sum_i \sum_j b_i B_j h_n^2 S(d_{n,i}, l_{n,j}) \\
&= \sum_i h_n (b_i S(d_{n,i}, \Lambda_{n,i}) + B_i S(\Delta_{n,i}, l_{n,i})) \\
&\quad + \sum_i \sum_j (-b_i A_{i,j} - B_j a_{j,i} + b_i B_j) h_n^2 S(d_{n,i}, l_{n,j}).
\end{aligned}
\tag{17}
$$

If $B_i = b_i$ and $b_i A_{i,j} + B_j a_{j,i} - b_i B_j = 0$, the change vanishes, i.e., the partitioned Runge–Kutta conserves a bilinear quantity $S$. Note that $b_i$ must not vanish because $A_{i,j} = B_j(1 - a_{j,i}/b_i)$. Therefore, the bilinear quantity $\lambda_n^\top \delta_n$ is conserved as

$$
\lambda_N^\top \delta_N = \lambda_n^\top \delta_n \text{ for } n = 0, \dots, N. \tag{18}
$$

Remark 3 indicates $\delta_n = \frac{\partial x_n}{\partial x_0}$. When $\lambda_N$ is set to $(\frac{\partial \mathcal{L}(x_N)}{\partial x_N})^\top$,

$$
\frac{\partial \mathcal{L}(x_N)}{\partial x_0} = \frac{\partial \mathcal{L}(x_N)}{\partial x_N} \frac{\partial x_N}{\partial x_0} = \lambda_N^\top \delta_N = \lambda_n^\top \delta_n = \frac{\partial \mathcal{L}(x_N)}{\partial x_n} \frac{\partial x_n}{\partial x_0}, \tag{19}
$$

Therefore, $\lambda_n = (\frac{\partial \mathcal{L}(x_N)}{\partial x_n})^\top$.

**Proof of Theorem 2:**  By solving the combination of the integrators in Eqs. (5) and (7), a change in a bilinear quantity $S(\delta, \lambda)$ that the continuous-time dynamics conserves is

$$
\begin{aligned}
S(\delta_{n+1}, \lambda_{n+1}) - S(\delta_n, \lambda_n) &= S(\delta_n + h_n \sum_i b_i d_{n,i}, \lambda_n + h_n \sum_i \tilde{b}_i l_{n,i}) - S(\delta_n, \lambda_n) \\
&= \sum_i b_i h_n S(d_{n,i}, \lambda_n) + \sum_i \tilde{b}_i h_n S(\delta_n, l_{n,i}) \\
&\quad + \sum_i \sum_j b_i \tilde{b}_j h_n^2 S(d_{n,i}, l_{n,j}) \\
&= \sum_{i \notin I_0} b_i h_n S(d_{n,i}, \Lambda_{n,i} - h_n \sum_j \tilde{b}_j(1 - a_{j,i}/b_i) l_{n,j}) \\
&\quad + \sum_i \tilde{b}_i h_n S(\Delta_{n,i} - h_n \sum_j a_{i,j} d_{n,j}, l_{n,i}) \\
&\quad + \sum_{i \notin I_0} \sum_j b_i \tilde{b}_j h_n^2 S(d_{n,i}, l_{n,j}) \\
&= \sum_{i \notin I_0} b_i h_n (S(d_{n,i}, \Lambda_{n,j}) + S(\Delta_{n,i}, l_{n,j})) \\
&\quad + \sum_{i \notin I_0} \sum_j (-b_i \tilde{b}_j(1 - a_{j,i}/b_i) - \tilde{b}_j a_{j,i} + b_i \tilde{b}_j) h_n^2 S(d_{n,i}, l_{n,j}) \\
&\quad + \sum_{i \in I_0} (\tilde{b}_i h_n S(\Delta_{n,i}, l_{n,j}) - \sum_j \tilde{b}_j a_{j,i} h_n^2 S(d_{n,i}, l_{n,j})) \\
&= \sum_{i \notin I_0} b_i h_n (S(d_{n,i}, \Lambda_{n,j}) + S(\Delta_{n,i}, l_{n,j})) \\
&\quad + \sum_{i \in I_0} h_n^2 (S(d_{n,i}, \Lambda_{n,j}) + S(\Delta_{n,i}, l_{n,j})) \\
&= 0.
\end{aligned}
\tag{20}
$$

Hence, the bilinear quantity $S(\delta, \lambda)$ is conserved.

**Proof of Remark 4:**  Eq. (6) can be rewritten as

$$
\begin{aligned}
\lambda_n &= \lambda_{n+1} - h_n \sum_{i=1}^s b_i l_{n,i} \\
l_{n,i} &= -\frac{\partial f}{\partial x}(X_{n,i}, t_n + c_i h_n)^\top \Lambda_{n,i}, \\
\Lambda_{n,i} &= \lambda_{n+1} - h_n \sum_{i=1}^s b_j \frac{a_{j,i}}{b_i} l_{n,j}.
\end{aligned}
\tag{21}
$$

Eq. (7) can be rewritten as

$$\lambda_n = \lambda_{n+1} - h_n \sum_{i=1}^{s} \tilde{b}_i l_{n,i},$$

$$l_{n,i} = -\frac{\partial f}{\partial x}(X_{n,i}, t_n + c_i h_n)^\top \Lambda_{n,i}, \tag{22}$$

$$\Lambda_{n,i} = \begin{cases} \lambda_{n+1} - h_n \sum_{j=1}^{s} \tilde{b}_j \frac{a_{j,i}}{b_i} l_{n,j} & \text{if } i \notin I_0 \\ -\sum_{j=1}^{s} \tilde{b}_j a_{j,i} l_{n,j} & \text{if } i \in I_0. \end{cases}$$

Because $a_{i,j} = 0$ for $j \geq i$, $a_{j,i} = 0$ for $j \leq i$. The intermediate adjoint variable $\Lambda_{n,i}$ is calculable from $i = s$ to $i = 1$ sequentially, i.e., the integration backward in time is explicit.

## C  Gradients in General Cases

### C.1  Gradient w.r.t. Parameters

For the parameter adjustment, one can consider the parameters $\theta$ as a part of the augmented state $\tilde{x} = [x \; \theta]^\top$ of the system

$$\frac{\mathrm{d}}{\mathrm{d}t}\tilde{x} = \tilde{f}(\tilde{x}, t), \; \tilde{f}(\tilde{x}, t) = \begin{bmatrix} f(x, t, \theta) \\ 0 \end{bmatrix}, \; \tilde{x}(0) = \begin{bmatrix} x_0 \\ \theta \end{bmatrix}. \tag{23}$$

The variational and adjoint variables are augmented in the same way. For the augmented adjoint variable $\tilde{\lambda} = [\lambda \; \lambda_\theta]^\top$, the augmented adjoint system is

$$\frac{\mathrm{d}}{\mathrm{d}t}\tilde{\lambda} = -\frac{\partial \tilde{f}}{\partial \tilde{x}}(\tilde{x}, t)^\top \tilde{\lambda} = -\begin{bmatrix} \frac{\partial f}{\partial x}^\top & 0 \\ \frac{\partial f}{\partial \theta}^\top & 0 \end{bmatrix} \begin{bmatrix} \lambda \\ \lambda_\theta \end{bmatrix} = \begin{bmatrix} -\frac{\partial f}{\partial x}^\top \lambda \\ -\frac{\partial f}{\partial \theta}^\top \lambda \end{bmatrix}. \tag{24}$$

Hence, the adjoint variable $\lambda$ for the system state $x$ is unchanged from Eq. (3), and the one $\lambda_\theta$ for the parameters $\theta$ depends on the former as

$$\frac{\mathrm{d}}{\mathrm{d}t}\lambda_\theta = -\frac{\partial f}{\partial \theta}(x, t, \theta)^\top \lambda, \tag{25}$$

and $\lambda_\theta(T) = (\frac{\partial \mathcal{L}(x(T), \theta)}{\partial \theta})^\top$.

### C.2  Gradient of Functional

When the solution $x(t)$ is evaluated by a functional $\mathcal{C}$ as

$$\mathcal{C}(x(t)) = \int_0^T \mathcal{L}(x(t), t)\mathrm{d}t, \tag{26}$$

the adjoint variable $\lambda_C$ that denotes the gradient $\lambda_C(t) = (\frac{\partial \mathcal{C}(x(T))}{\partial x(t)})^\top$ of the functional $\mathcal{C}$ is given by

$$\frac{\mathrm{d}}{\mathrm{d}t}\lambda_C = -\frac{\partial f}{\partial x}(x, t)^\top \lambda_C + \frac{\partial \mathcal{L}(x(t), t)}{\partial x(t)}, \; \lambda_C(T) = \mathbf{0}. \tag{27}$$

## D  Implementation Details

### D.1  Robustness to Rounding Error

By definition, the naive backpropagation algorithm, baseline scheme, ACA, and the proposed symplectic adjoint method provide the exact gradient up to rounding error. However, the naive backpropagation algorithm and baseline scheme obtained slightly worse results on the GAS, POWER, and HEPMASS datasets. Due to the repeated use of the neural network, each method accumulates the gradient of the parameters $\theta$ for each use. Let $\theta_{n,i}$ denote the parameters used in the $i$-th stage of $n$-th

Table A1: Results on learning physical systems without the deterministic option.

| | KdV Equation | | | Cahn–Hilliard System | | |
|---|---|---|---|---|---|---|
| | MSE $(\times 10^{-3})$ | mem. | time | MSE $(\times 10^{-6})$ | mem. | time |
| adjoint method [2] | 1.61±3.23 | **181.4**± 0.0 | 240±16 | 5.58±2.12 | **181.4**± 0.0 | 805±25 |
| backpropagation [2] | 1.61±3.24 | 733.9±15.6 | 94± 4 | 5.45±1.55 | 3053.5±22.9 | 382±11 |
| ACA [46] | 1.61±3.24 | 734.5±20.3 | 120± 4 | 6.00±3.27 | 780.4±22.9 | 422±16 |
| proposed | 1.61±3.58 | 182.1± 0.0 | 141± 7 | 5.48±1.90 | 182.1± 0.0 | 480±19 |

Mean-squared errors (MSEs) in long-term predictions, peak memory consumption [MiB],
and computation time per iteration [ms/itr].

step even though their values are unchanged. The backpropagation algorithm obtains the gradient $\frac{\partial \mathcal{L}}{\partial \theta}$
with respect to the parameters $\theta$ by accumulating the gradient over all stages and steps one-by-one as

$$\frac{\partial \mathcal{L}}{\partial \theta} = \sum_{\substack{n=0,\ldots,N-1, \\ i=1,\ldots,s}} \frac{\partial \mathcal{L}}{\partial \theta_{n,i}}. \tag{28}$$

When the step size $h_n$ at the $n$-th step is sufficiently small, the gradient $\frac{\partial \mathcal{L}}{\partial \theta_{n,i}}$ at the $i$-th stage may be
insignificant compared with the accumulated gradient and be rounded off during the accumulation.

Conversely, ACA accumulates the gradient within a step and then over time steps; this process can be
expressed informally as

$$\frac{\partial \mathcal{L}}{\partial \theta} = \sum_{n=0}^{N-1} \left( \sum_{i=1}^{s} \frac{\partial \mathcal{L}}{\partial \theta_{n,i}} \right). \tag{29}$$

Further, according to Eqs. (6) and (25), the (symplectic) adjoint method accumulates the adjoint
variable $\lambda$ (i.e., the transpose of the gradient) within a step and then over time steps as

$$\lambda_{\theta,n} = \lambda_{\theta,n+1} - h_n \left( \sum_{i=1}^{s} B_i \left( -\frac{\partial f}{\partial \theta_{n,i}}(X_{n,i}, t + C_i h_n, \theta_{n,i})^\top \Lambda_{n,i} \right) \right). \tag{30}$$

In these cases, even when the step size $h_n$ at the $n$-th step is small, the gradient summed within a step
(over $s$ stages) may still be significant and robust to rounding errors. This is the reason the adjoint
method, ACA, and the symplectic adjoint method performed better than the naive backpropagation
algorithm and baseline scheme for some datasets. Note that this approach requires additional memory
consumption to store the gradient summed within a step, and it is applicable to the backpropagation
algorithm with a slight modification.

## D.2 Memory Consumption Optimization

Following Eqs. (21) and (22), a naive implementation of the adjoint method retains the adjoint
variables $\Lambda_{n,i}$ at all stages $i = 1, \ldots, s$ to obtain their time-derivatives $l_{n,i}$, and then adds them up to
obtain the adjoint variable $\lambda_n$ at the $n$-th time step. However, as Eq. (25) shows, the adjoint variable
$\lambda_\theta$ for the parameters $\theta$ is not used for obtaining its time-derivative $\frac{\mathrm{d}}{\mathrm{d}t}\lambda_\theta$. One can add up the adjoint
variable $\Lambda_{\theta n,i}$ for the parameters $\theta$ at stage $i$ one-by-one without retaining it, thereby reducing the
memory consumption proportionally to the number of parameters *times* the number of stages. A
similar optimization is applicable to the adjoint method.

## D.3 Parallelization

The memory consumption and computation time depend highly on the implementations and devices.
Being implemented on a GPU, the convolution operation can be easily parallelized in space and
exhibits a non-deterministic behavior. To avoid the non-deterministic behavior, PyTorch provides
an option TORCH.BACKENDS.CUDNN.DETERMINISTIC, which was used to obtain the results in
Section 5.2, following the original implementation [31]. Without this option, the memory con-
sumption increased by a certain amount, and the computation times reduced due to the aggressive

parallelization, as shown by the results in Table A1. Even then, the proposed symplectic adjoint method occupied the smallest memory among the methods for the exact gradient. The increase in the memory consumption is proportional to the width of a neural network; therefore, it is negligible when the neural network is sufficiently deep.

Note that the results in Section 5.1 were obtained without the deterministic option.

Table A2: Results obtained for continuous normalizing flows.

| | MINIBOONE ($M=1$) | | | GAS ($M=5$) | | | POWER ($M=5$) | | |
|---|---|---|---|---|---|---|---|---|---|
| | NLL | mem. | time | NLL | mem. | time | NLL | mem. | time |
| adjoint method [2] | 10.59±0.17 | 170±0 | 0.74±0.04 | -10.53±0.25 | 24±0 | 4.82±0.29 | -0.31±0.01 | **8.1**±0.0 | 6.33±0.18 |
| backpropagation [2] | 10.54±0.18 | 4,436±115 | 0.91±0.05 | -9.53±0.42 | 4,479±250 | 12.00±0.93 | -0.24±0.05 | 1710.9±193.1 | 10.64±2.73 |
| baseline scheme | 10.54±0.18 | 4,457±115 | 1.10±0.04 | -9.53±0.42 | 1,858±228 | 5.48±0.25 | -0.24±0.05 | 515.2±122.0 | 4.37±0.70 |
| ACA [46] | 10.57±0.30 | 306±0 | 0.77±0.02 | -10.65±0.45 | 73±0 | 3.98±0.14 | -0.31±0.02 | 29.5±0.5 | 5.08±0.88 |
| proposed | 10.49±0.11 | **95**±0 | 0.84±0.03 | -10.89±0.11 | **20**±0 | 4.39±0.23 | -0.31±0.02 | 9.2±0.0 | 5.73±0.43 |

| | HEPMASS ($M=10$) | | | BSDS300 ($M=2$) | | | MNIST ($M=6$) | | |
|---|---|---|---|---|---|---|---|---|---|
| | NLL | mem. | time | NLL | mem. | time | NLL | mem. | time |
| adjoint method [2] | 16.49±0.25 | 40±0 | 4.19±0.15 | -152.04±0.09 | 577±0 | 11.70±0.44 | 0.918±0.011 | 1,086±4 | 10.12±0.88 |
| backpropagation [2] | 17.03±0.22 | 5,254±137 | 11.82±1.33 | — | — | — | — | — | — |
| baseline scheme | 17.03±0.22 | 1,102±174 | 4.40±0.40 | — | — | — | — | — | — |
| ACA [46] | 16.41±0.39 | 88±0 | 3.67±0.12 | -151.27±0.47 | 757±1 | 6.97±0.25 | 0.919±0.003 | 4,332±1 | 7.94±0.63 |
| proposed | 16.48±0.20 | **35**±0 | 4.15±0.13 | -151.17±0.15 | **283**±2 | 8.07±0.72 | 0.917±0.002 | **1,079**±1 | 9.42±0.32 |

Negative log-likelihoods (NLL), peak memory consumption [MiB], and computation time per iteration [s/itr]. The medians ± standard deviations of three runs.