# OpenReview forum: "Symplectic Adjoint Method for Exact Gradient of Neural ODE with Minimal Memory"
_NeurIPS.cc/2021/Conference — NeurIPS 2021 Poster_

### Official Review · Reviewer_1e2r · 2021-07-14

**Rating:** 7
**Confidence:** 2

**Summary:**

The paper proposes a new algorithm for propagating gradients through neural
ODEs, which minimizes peak memory requirements by checkpointing not only the
individual solver steps but also the internal stages. The considered, specific,
partitioned Runge--Kutta method has symplectic properties and provides exact
gradients. The experiments demonstrate the reduced peak memory cost together
with competitive runtimes.


**Limitations And Societal Impact:**

The authors mentioned both in the checklist and in the paper that the
method is only applicable to Runge--Kutta methods which is a fair and valid
limitation. The other mentioned limitation relates to the runtimes: Table 1
shows the largest estimate of computational cost for the proposed method. The
experiments do however not reflect this limitation. The first paragraph of page
8 mentions this observation and refers to possible memory-related issues. A
possible additional experiment could therefore be in computational environments
with an abundance of memory, to observe the validity of the estimated, increased
computational cost (but I do not see this as a strict requirement for this paper).


**Main Review:**

__Originality__
To the best of my knowledge, the proposed method for continuous sensitivity
analysis appears to be novel. Previous approaches for adjoint computation of
neural ODEs are properly cited and compared to.
Large parts of the specific algorithm seem to have been developed in [36]. To
best judge on the differences and on the specific contribution of this paper, an
explicit comparison to [36] would be very helpful.

__Quality__
The submission appears technically sound. It includes an adequate discussion of
related methods for sensitivity analysis, and a thorough derivation of the
proposed symplectic adjoint method. Proofs are included or referred to.


__Clarity__
The submission is well-structured and clearly written, and seems to contain
sufficient information for reimplementing the method. The added code further
helps reproducibility. More specific comments for improvement can be found
below.

__Significance__
The proposed method provides a new algorithm for computing continuous
sensitivities with reduced memory-cost, and thereby provides a significant
contribution. I believe there is the potential for the proposed method to be
used, or at least considered, in future research or application of NODEs in
low-memory scenarios.


__Questions for the authors__
- To the best of my knowledge, symplectic ODE solvers rely on fixed steps (see
  e.g. [16]). Could you clarify how your method is able to benefit from
  step-size adaptation while still being symplectic?

  Similarly, my understanding of symplectic ODE solvers is that they are able to
  _exactly_ preserve an _approximate_ Hamiltonian, which is not equal to the true,
  continuous Hamiltonian. This would contradict your claim on computing the
  _exact_ gradient. Could you clarify my possible misconception?
- It would be nice to include two other adjoint methods into the discussion:
  (i) the CVODES adjoint described in [2] uses interpolation of the forward
  solution, and (ii) the quadrature-based adjoint computation described in [3]
  employs numerical quadrature instead of an ODE solver. Both are also described
  and used in [4]. Discussing their respective advantages and limitations would
  provide helpful information (though I am aware that [3] and [4] are only
  available as preprints at the current time such that their inclusion is not
  strictly necessary).


__Additional, minor comments and questions__
- Specifically mentioning _"peak"_ memory cost could be helpful; e.g. in 2.2 I
  was initially confused about the memory cost: M components, with O(NsL) memory
  cost, initially suggested a total O(MNsL) cost (instead of O(M+NsL)). Being
  reminded about "peak" memory usage clarified this misunderstanding for me.
- Introduction: l. 34-35 would benefit from a citation to support the claim.
- l. 55: "Backprop tends to suffer from rounding errors" - I was not able to
  find a comment on this in [41]. However, the information provided in the
  supplementary D.1 seems reasonable.
- 2.1: "Neural ODE components" is used throughout the paper, but never
  introduced. An introducing sentence could help the clarity
- 2.1: Is the number of layers "L" really helpful for the discussion? All
  compared methods scale the same way in "L", and only the discussion regarding
  M, N, s seems relevant.
- l. 71: It seems misleading to claim that the original study on neural ODE
  "introduced" the adjoint method - they rather "used" this method. Multiple
  subsequently cited papers predate [2] and reflect that statement.
- l. 81-82: Why should the size of the adjoint variable lead to numerical
  issues? An explanation, or a supporting citation, would be helpful.
- Throughout the paper the "Dormand--Prince" method is mentioned, but to the
  best of my knowledge there are multiple methods developed by and named after
  Dormand and Prince - as you also mention in your experiments. Specifying the
  specific method (i.e. "fifth-order Dormand--Prince", "DOPRI5", or possibly
  even "Runge--Kutta 4/5")
- l. 104-105: The step-size and computational cost depends on the specified
  tolerance level - for high tolerances, low orders could be preferrable.
- l. 105-106: High-order symplectic integrators do exist. See e.g. [1].
- l. 127: "automatic differentiation" seems misleading - continuous adjoint
  sensitivities are precisely the alternative to automatic differentiation.
- Remark 3: Could you clarify further what "is _automatically_ solved by the same
  Runge--Kutta method" means?
- l. 261: It seems strong to state robustness to rounding errors; instead,
  becoming "more robust" is already a significant improvement.
- Experiments: Doing more than three runs would be helpful to better assess the
  standard deviation.
- Experiments "Different Runge--Kutta Methods": Please specify the used
  tolerance levels. In this context, I would imagine that the performance of the
  solvers strongly depends on the tolerance level; for lower tolerances, DOPRI8
  might provide the best performance.
- Supplementary, eq. (12), typo: t -> T


[1] "Composition constants for raising the orders of unconventional schemes for
ordinary differential equations", Kahan and Li (1997).

[2] "SUNDIALS: Suite of Nonlinear and Differential/Algebraic Equation Solvers",
Hindmarsh et al. (2005)

[3] "Stiff Neural Ordinary Differential Equations", Kim et al. (2021)

[4] "Universal Differential Equations for Scientific Machine Learning",
Rackauckas et al. (2020)


**Time Spent Reviewing:**

6

---

> ### Author Response · Authors · 2021-08-10
> **Response to Reviewer 1e2r**
>
> We would like to thank the reviewer for careful reading and valuable comments. We are encouraged that the reviewer found that "the proposed method for continuous sensitivity analysis appears to be novel." and that the reviewer recognized the submission as "technically sound" and "well-structured and clearly written". We have addressed the main comments as below.
>
> **Originality**
>
> > To best judge on the differences and on the specific contribution of this paper, an explicit comparison to [36] would be very helpful.
>
> The reference [36] is a purely theoretical paper, and a naive implementation of the algorithm in [36] consumes the same memory and computation cost as the backpropagation (line 191-193). The novelty of this study is the proposal of the implementation scheme that uses appropriate checkpoints and reduces the memory consumption, proposed in Section 4.3 and in Algorithms 1 and 2.
>
> **Questions for the authors**
>
> > To the best of my knowledge, symplectic ODE solvers rely on fixed steps (see e.g. [16]). Could you clarify how your method is able to benefit from step-size adaptation while still being symplectic? Similarly, my understanding of symplectic ODE solvers is that they are able to exactly preserve an approximate Hamiltonian, which is not equal to the true, continuous Hamiltonian. This would contradict your claim on computing the exact gradient. Could you clarify my possible misconception?
>
> Thank you for your insightful comment. For obtaining the exact gradient, the true or approximate Hamiltonian is not necessarily preserved, but only the bilinear quantity $S$ of the adjoint variable $\lambda$ and variational variable $\delta$ should be preserved (Theorem 1). Any symplectic ODE solvers are known to preserve this kind of bilinear quantity [36]. With a variable step size, the symplectic ODE solvers no longer preserve the approximate Hamiltonian, but still preserve the bilinear quantity $S$ exactly. The symplecticity is not a necessary condition but a sufficient condition (line 182), and one can relax the condition (i.e., change the step size) for obtaining the exact gradient. We will add this important discussion in the final version.
>
> > It would be nice to include two other adjoint methods into the discussion: (i) the CVODES adjoint described in [2] uses interpolation of the forward solution, and (ii) the quadrature-based adjoint computation described in [3] employs numerical quadrature instead of an ODE solver. Both are also described and used in [4]. Discussing their respective advantages and limitations would provide helpful information (though I am aware that [3] and [4] are only available as preprints at the current time such that their inclusion is not strictly necessary).
>
> In practice, the best integrator for the forward integration depends on the targeted system, and the best integrator for the adjoint method does the same. SUNDIALS in [2] and SciML in [4] are packages that provide many integrators and can choose the best ones. For example, CVODE and CVODES provide interpolating functions (including quadrature methods) for integrating an ODE or its adjoint system. A recent study [3] has demonstrated that, when used for a neural ODE solving stiff equations, quadrature methods for the adjoint system can reduce the computation cost in exchange for the additional memory consumption. In contrast, the symplectic adjoint method is assumed to be used with Runge-Kutta methods. In the future, we will provide the symplectic adjoint method as a part of such packages and make it available for appropriate systems. At present, we will revise the manuscript to add this discussion.
>
> **Additional, minor comments and questions**
>
> > Specifically mentioning "peak" memory cost could be helpful; e.g. in 2.2 I was initially confused about the memory cost: M components, with O(NsL) memory cost, initially suggested a total O(MNsL) cost (instead of O(M+NsL)). Being reminded about "peak" memory usage clarified this misunderstanding for me.
>
> Sorry for the confusion. The theoretical and experimental memory costs summarized in Tables 1--4 and A2 are peak memory costs. In Section 2.2, ANODE stores and recomputes each NODE component separately, and hence, consumes the memory of $O(M)$ for checkpoints of $M$ components and the memory of $O(NsL)$ to store the single NODE component that ANODE currently performs the backpropagation; the memory consumption is $O(M+NsL)$ in total. We will revise the sentence to avoid the confusion.
>
> > 2.1: Is the number of layers "L" really helpful for the discussion? All compared methods scale the same way in "L", and only the discussion regarding M, N, s seems relevant.
>
> We aimed to emphasize that the memory for checkpoints ($O(MN+s)$) is much smaller than that for the backpropagation of a single use of an $L$-layered neural network ($O(L)$) in practice. When using $L$, one can say that in fact, $O(L)$ has an extremely large coefficient compared with $O(MN+s)$. Moreover, ACA consumes the memory of $O(sL)$ for the backpropagation; using $L$ makes it easier to understand the memory bottleneck of ACA.
>
> > l. 105-106: High-order symplectic integrators do exist. See e.g. [1].
>
> Sorry for the confusion. We meant that, while higher-order symplectic time-reversal integrators do exist, they cannot be used in place of MALI. MALI obtains the consistent state $x$ in the forward and backward integrations and thereby suppresses the memory consumption. This is because MALI employs a symplectic time-reversal integrator (asynchronous leapfrog integrator). However, higher-order symplectic time-reversal Runge-Kutta methods are implicit methods [Section V, 16]. An implicit method performs a root-finding algorithm for each step and is extremely computationally expensive. Hence, it cannot be used in place of MALI. Higher-order explicit time-reversal *partitioned* Runge-Kutta methods exist and can be used in place of MALI; however, they are inapplicable to physics systems or dynamical systems without velocity. Methods in [1] are compositions of time-reversal methods, each of which is implicit (i.e., computationally expensive) or partitioned (i.e., applicable only to limited tasks).
>
> > Experiments: Doing more than three runs would be helpful to better assess the standard deviation.
>
> We followed the original study of FFJORD. Because we used the same random seeds for all methods and each method obtains the exact gradient up to rounding or numerical error, each method obtained almost the same results. Hence, we consider that three runs are enough to confirm the difference. We obtained 15 runs for physics systems.
>
> > Experiments "Different Runge--Kutta Methods": Please specify the used tolerance levels. In this context, I would imagine that the performance of the solvers strongly depends on the tolerance level; for lower tolerances, DOPRI8 might provide the best performance.
>
> As mentioned in line 240, we set the absolute and relative tolerances to atol=$10^{−8}$ and rtol=$10^{−6}$, respectively, following the original experiment. As you suggest, a different solver would be the best with different tolerance value. However, our aim is to compare the methods to obtain gradient in varying problem settings, but not solvers. The important point is that the DOPRI5 is the best choice in this case, and the symplectic adjoint method works faster and consumes less memory than the adjoint method when using the best solver.
>
> > l. 81-82: Why should the size of the adjoint variable lead to numerical issues? An explanation, or a supporting citation, would be helpful.
>
> Sorry for the inappropriate explanation. We meant that, with more parameters, the probability that at least one parameter does not satisfy the tolerance value is increased, and the expected number of steps is increased. We will revise the explanation.
>
> The remaining comments are on the improvement of the descriptions such as adding a supportive reference (already cited by another sentence) and fixing a typo. We thank the reviewer for the detailed suggestions and swear to revise the manuscript following them.

---

> > ### Comment · Reviewer_1e2r · 2021-08-11
> > **Update after Author Response**
> >
> > Thank you for the thorough, clarifying response. I raised my score to indicate my recommendation to accept this paper.

---

> > > ### Author Response · Authors · 2021-08-11
> > > **Acknowledgment**
> > >
> > > We are pleased that our response has resolved your concerns. We deeply appreciate the update.

---

### Official Review · Reviewer_rdkK · 2021-07-16

**Rating:** 5
**Confidence:** 4

**Summary:**

In this paper, the authors propose a new adjoint method for training neural ODEs, named the symplectic adjoint method, which is an adjoint method solved by a symplectic integrator. The proposed symplectic adjoint method obtains the exact gradient with memory proportional to the number of uses plus the network size. Numerically, the authors show that the symplectic adjoint method consumes much less memory than the naive backpropagation algorithm and checkpointing schemes, and performs faster than the adjoint method, and is robust to round off errors.

**Main Review:**

Strength:
1. Propose a new numerical algorithm for the backpropagation of NODEs.

2. Runge-Kutta only (which is generally enough), so accessible to higher-order methods. It has more memory advantage for higher-order methods.

3. Comprehensive numerical experiments and nice performance.

Weakness:
1. The memory consumption is proportional to the number of time steps. When the ODE is not smooth enough for higher-order methods and/or integrating for a long time period, this may cause an issue. Please address this point and I am happy to raise the rating.

**Time Spent Reviewing:**

4 hours

---

> ### Author Response · Authors · 2021-08-10
> **Response to Reviewer rdkK**
>
> We would like to thank the reviewer for valuable comments. We have addressed the concern as below.
>
> > The memory consumption is proportional to the number of time steps. When the ODE is not smooth enough for higher-order methods and/or integrating for a long time period, this may cause an issue. Please address this point and I am happy to raise the rating.
>
> In practice, the memory for checkpoints does not cause an issue. The Cahn-Hilliard system in Section 5.2 is known to be a stiff (less smooth) system [9], and it requires roughly 10 checkpoints for the forward integration. FFJORD for MNIST in Section 5.1 requires more than 100 checkpoints. Even then, the symplectic adjoint method consumes the memory smaller than or at similar level to the adjoint method (Tables 2, 4, and A1).
>
> We performed an additional experiment using FFJORD for MNIST, where we increased the number of checkpoints by reducing the tolerance, as summarized below. Hundreds of checkpoints increased the memory consumption only little.
>
> | method                    |  tolerance | # checkpoints |  memory |
> |:--------------------------|-----------:|--------------:|--------:|
> | adjoint method            |  $10^{-5}$ |            10 | 1083 MB |
> | symplectic adjoint method |  $10^{-5}$ |           107 | 1079 MB |
> |                           |  $10^{-8}$ |           370 | 1115 MB |
> |                           |  $10^{-9}$ |           582 | 1151 MB |
> |                           | $10^{-10}$ |          2591 | 1412 MB |
>
> These facts indicate that the memory of $O(MN+s)$ for checkpoints is almost negligible compared with that of $O(L)$ for the backpropagation of a single use of an $L$-layered neural network. In other words, $O(L)$ has a large coefficient compared with $O(MN+s)$. Only when using an unreasonably lower-order method and a small neural network, the checkpoints consume a non-negligible memory (second or third-order method in Table 3).
>
> When the target ODE is so stiff that explicit Runge-Kutta methods need much more steps, implicit methods are often used. Implicit methods are out of scope of most related works. The symplectic adjoint method is still applicable to implicit Runge-Kutta methods theoretically, but the current implementation does not consider them.
>
> We believe that these results resolve the reviewer's concern.

---

### Official Review · Reviewer_G97n · 2021-07-18

**Rating:** 6
**Confidence:** 4

**Summary:**

This paper proposed to view the state and its adjoint in Neural ODEs as a partitioned system, hence the partitioned Runge-Kutta method can be applied, hence the integrator is symplectic and is accurate in terms of gradient estimation. To my knowledge, this is the first time that the idea of the higher-order symplectic integrator is introduced to Neural ODE.

**Limitations And Societal Impact:**

Yes

**Main Review:**

Pros: \
The idea of a symplectic Runge-Kutta solver and view the adjoint system as a partitioned system is well-studied in the numerical analysis community, but to my knowledge, it's not systematically introduced into the deep learning community. The paper is in general well written with solid theoretical analysis.

Comments: \
I would suggest the authors remove the "minimal memory" in the title since the memory cost still grows linearly with integration time steps, while the original adjoint method and MALI all have a constant memory w.r.t integration time.

I'm curious how does the proposed method compare with MALI which has a constant $O(1)$ memory cost and is lower than symplectic-adjoint ($O(T)$ memory). MALI is a second-order method, which hinders the numerical accuracy of high-order solvers. But from my experience, even a 2nd order solver is sufficient for most deep learning with Neural ODEs. It would be more convincing if the authors could discuss or validate in experiments.

Another suggestion is that the authors might want to expand a bit more on why ACA uses $sL$ memory while symplectic-adjoint uses $L$ memory during backward, current writing does not explain it sufficiently and it's a bit hard to understand at first reading. For an RK-step of $s$ stages, ACA performs backward through all $s$ stages at once, while symplectic-adjoint performs backward pass for each stage of the $s$ stages, and each stage can be deleted from the memory once it's computed. This depends on the result of Thm 1, however, I'm curious if Thm1 always holds for any ODE system? Does it require some properties of the ODE system such as smoothness? Can it be applied to some neural-jump ODE or stochastic ODE?

**Time Spent Reviewing:**

3

---

> ### Author Response · Authors · 2021-08-10
> **Response to Reviewer G97n**
>
> We would like to thank the reviewer for careful reading and insightful comments. We are glad that the reviewer found that "the paper is in general well written with solid theoretical analysis." We have addressed all the comments as below.
>
> **Pros**
>
> > to my knowledge, it's not systematically introduced into the deep learning community.
>
> Yes, this is the first time to introduce the idea of a symplectic Runge-Kutta solver for adjoint method into the deep learning community. Moreover, a naive implementation of the idea requires the same memory consumption and computation cost as the backpropagation (line 191-193). The novelty of this study is the proposal of the implementation scheme that uses appropriate checkpoints and reduces the memory consumption, proposed in Section 4.3 and in Algorithms 1 and 2.
>
> **Comments**
>
> > I would suggest the authors remove the "minimal memory" in the title
>
> Thank you for the suggestion. We will change the title to avoid the confusion.
>
> > I'm curious how does the proposed method compare with MALI which has a constant $O(1)$ memory cost and is lower than symplectic-adjoint ($O(T)$ memory).
>
> We emphasize that MALI, the adjoint method, and the symplectic adjoint method consume the memory of $O(L)$ for the backpropagation of a single use of an $L$-layered neural network, while most previous works omitted $L$. Then, roughly speaking, $O(L)\simeq O(T+L)$. For the forward integration, the Cahn-Hilliard system in Section 5.2 requires roughly 10 checkpoints, and FFJORD for MNIST in Section 5.1 requires more than 100 checkpoints. Even then, the symplectic adjoint method consumes almost the same memory as the adjoint method (Tables 2, 4, and A1).
>
> We performed an additional experiment using FFJORD for MNIST, where we increased the number of checkpoints by reducing the tolerance, as summarized below. Hundreds of checkpoints increased the memory consumption only little.
>
> |  tolerance | # checkpoints |  memory |
> |-----------:|--------------:|--------:|
> |  (adjoint) |            10 | 1083 MB |
> |  $10^{-5}$ |           107 | 1079 MB |
> |  $10^{-8}$ |           370 | 1115 MB |
> |  $10^{-9}$ |           582 | 1151 MB |
> | $10^{-10}$ |         2,591 | 1412 MB |
>
> These facts indicate that the memory of $O(MN+s)$ for checkpoints is almost negligible compared with that of $O(L)$ for the backpropagation. Only when using an unreasonably lower-order method and a small neural network, the checkpoints consume a non-negligible memory (second or third-order method in Table 3).
>
> > from my experience, even a 2nd order solver is sufficient for most deep learning with Neural ODEs. It would be more convincing if the authors could discuss or validate in experiments.
>
> A second-order solver is insufficient for continuous-time tasks. We examined the second-order adaptive Heun method for continuous normalizing flow, FFJORD (Table 3). The adaptive Heun method requires *50 times longer computation time* than the fifth-order Dormand-Prince method to satisfies the same tolerance. MALI is expected to require even longer computation time because it is a second-order method without adaptive time-stepping. The same can happen to other continuous-time tasks needing the tolerance such as time-series analysis [22] and modeling dynamical systems [32][37]. On the other hand, the symplectic adjoint method is generally applicable to any Runge-Kutta methods of arbitrary orders. Moreover, learning physics systems has become a hot topic in deep learning community [16][27], which is also defined in continuous time. Therefore, we conclude that a second-order method is practically sufficient only for classical tasks such as image recognition, for which classical deep learning works well.
>
> > Another suggestion is that the authors might want to expand a bit more on why ACA uses $sL$ memory while symplectic-adjoint uses $L$ memory during backward, current writing does not explain it sufficiently and it's a bit hard to understand at first reading. For an RK-step of $s$ stages, ACA performs backward through all stages at once, while symplectic-adjoint performs backward pass for each stage of the $s$ stages, and each stage can be deleted from the memory once it's computed.
>
> Yes, your understanding is correct. We will improve the explanation (2nd paragraph of Section 4.3) following your valuable comment.
>
> > I'm curious if Thm1 always holds for any ODE system? Does it require some properties of the ODE system such as smoothness? Can it be applied to some neural-jump ODE or stochastic ODE?
>
> Thm1 always holds for any ODE systems because Thm1 is a property of Runge-Kutta methods, and not a property of target ODE systems.
>
> A neural-jump ODE can be treated as ODEs with the discontinuity inferred using other neural networks [Herrera et al.]. Both the adjoint method and the symplectic adjoint method are applicable to the ODEs, and the backpropagation is used for the remaining part. For stochastic ODE, a modified version of the adjoint method is applicable [Li et al.], and the symplectic adjoint method may also be applicable when the same modification is applied.
>
> We will revise the descriptions to improve the clarity and add potential future works.
>
> - [Herrera et al.] Herrera et al., "Neural Jump Ordinary Differential Equations: Consistent Continuous-Time Prediction and Filtering," International Conference on Learning Representations (ICLR), 2021.
> - [Li et al.] Li et al., "Scalable Gradients for Stochastic Differential Equations," International Conference on Artificial Intelligence and Statistics (AISTATS), 2020.

---

> > ### Comment · Reviewer_G97n · 2021-08-17
> > **Updated response**
> >
> > Thanks for your response, most of my concerns are addressed,but I still have a few concerns.
> >
> > 1 Regarding the memory in updated table.
> >
> > I  don't think it's completely correct to claim the $O(MN+s)$ memory is negligible compared to the $O(L)$ memory. For a flow model, the dimension of hidden state does not change, e.g. $dim(z) = dim(f(z,t))$. However, this does not imply $f(z,t)$ takes a small memory, $f$ can be constructed as mapping to a very high dimension in the middle layer, then mapping back to $dim(z)$. In this case, the memory by just evaluating $f(z,t)$ will dominate.
> >
> > If I remember correctly, ffjord is a quite large model in terms of $f$, while the $dim(z)$ for MNIST is a small value (28x28x1). Hence I think you would observe a big increase in memory, if $f$ is small (e.g. 1-layer) but $z$ is large (e.g. ImageNet image size is 256x256x3). I think it's not a problem unique to your method, but it's a draw back of all existing checkpoint strategy (which uses more checkpoints with longer time). The only two methods not suffering this problem are adjoint and MALI.
> >
> > 2 Regarding second-order solvers ( just discussion, I don't think this point would affect my rating)
> >
> > I agree that for model-driven learning second-order solver is insufficient, but for most data-driven learning with over-parameterized models, such a high accuracy is not always necessary, because over-parameterized models would behave like an ensemble and the errors would cancel pretty much. Also MALI is an adaptive solver in the default implementation, if I remember correctly.

---

> > > ### Author Response · Authors · 2021-08-17
> > > **Response to Reviewer G97n : No Free Lunch, but Certainly Useful in Practice**
> > >
> > > We are glad that most of your concerns are addressed.
> > >
> > > **comment 1**
> > >
> > > > $f$ can be constructed as mapping to a very high dimension in the middle layer, then mapping back to $dim(z)$. In this case, the memory by just evaluating $f(z,t)$ will dominate.
> > >
> > > > ffjord is a quite large model in terms of, while the for MNIST is a small value (28x28x1).
> > >
> > > Yes, and it is often the case. A large model is necessary to obtain a good log-likelihood even for a small-sized sample. This is true for tabular datasets.
> > >
> > > FFJORD uses convolution layers for an image. Then, the middle layer has the same size as the input and checkpoints, while their number of channels may differ. Even if the image size increases, the memory for the backpropagation of $f$ and that for checkpoints increase at the same rate, and the former keeps dominating.
> > >
> > > > I think you would observe a big increase in memory, if $f$ is small (e.g. 1-layer) but is large (e.g. ImageNet image size is 256x256x3). I think it's not a problem unique to your method, but it's a draw back of all existing checkpoint strategy (which uses more checkpoints with longer time). The only two methods not suffering this problem are adjoint and MALI.
> > >
> > > We respectfully consider that this is *an extreme and non-practical case*. A 1-layer neural network does not have the universal approximation property and must not work for ImageNet.
> > >
> > > Like FFJORD in Section 5.1, a larger neural network is necessary for a larger-sized sample. Like physics simulation in Section 5.2, a smaller neural network is enough for a smaller-sized sample (we used only two hidden layers). In both *practical cases*, the memory for checkpoints was negligible compared with that for backpropagation.
> > >
> > > **comment 2**
> > >
> > > > I agree that for model-driven learning second-order solver is insufficient, but for most data-driven learning with over-parameterized models, such a high accuracy is not always necessary, because over-parameterized models would behave like an ensemble and the errors would cancel pretty much.
> > >
> > > If a high accuracy is not necessary, the number of steps is small and the memory for checkpoints is negligible. Also, with an over-parameterized model, the memory for checkpoints is negligible compared with the memory for the backpropagation, which is proportional to the model size. In both cases, the symplectic adjoint method works similarly to MALI. For a high accuracy, MALI needs less memory and much longer computation time than the symplectic adjoint method; there is a trade-off.
> > >
> > > The original study of FFJORD thoroughly examined the hyperparameters of the model size (number of layers and width) and the tolerance, and confirmed that a large neural network did not always achieve the better results. Hence, a large neural network does not always cancel the errors.
> > >
> > > > Also MALI is an adaptive solver in the default implementation, if I remember correctly.
> > >
> > > Thank you for correction. Yes, MALI used an adaptive solver. Even then, MALI is expected to require tens times longer computation time for tasks that need a high accuracy, like the adaptive Heun method as shown in Table 3.
> > >
> > > **summary**
> > >
> > > As discussed above, the memory for checkpoints is not problematic except for an extreme and non-practical case.
> > >
> > > Your Comment 1 focuses on a small model, and your comment 2 focuses on a large model. Following the no free lunch theorem, a method cannot be the best choice for both cases. We do not claim that the symplectic adjoint method is a universal method. We agree that there exist many tasks for which MALI is the best choice (e.g., image recognition), but MALI is not a universal method, too (e.g., inapplicable to some dynamical systems, and long computation time for a high accuracy).
> > >
> > > As we replied to Reviewer 1e2r, a practical package for numerical integration such as SUNDIALS provides many methods for obtaining the gradient, and chooses the best one depending on given tasks and resources. We demonstrated in our experiments that the symplectic adjoint method has practical advantages in many criteria (not in all criteria); hence, the symplectic adjoint method is worth providing for many practical situations where it is the best choice.
> > >
> > > In the final version, we will respectfully introduce the related works (adjoint method and MALI), their advantages, and their trade-off with the symplectic adjoint method in details.
> > >
> > > We hope that this discussion resolves your remaining concerns.

---

> > > > ### Comment · Reviewer_G97n · 2021-08-17
> > > > **Response to authors**
> > > >
> > > > Thanks for your response. I still don't agree on your response, but these discussions don't affect your novelty, it only affects your discussion in paper, so I won't lower rating based on the following discussions.
> > > >
> > > > > Your Comment 1 focuses on a small model
> > > >
> > > > To be clear, my point is, when integration time is long enough (compared to $f$), you will observe a significant increase in memory. Checkpointing is memory-consuming. Long-time integration is often necessary in practice, and you can't call it "non-practical".
> > > >
> > > > > MALI needs less memory and much longer computation time than the symplectic adjoint method; there is a trade-off.
> > > >
> > > > High accuracy in integration is different from high-accuracy of model performance. At least when I run FFJORD with MALI, I don't observe much increase in running time compared to adjoint, and the model performance is better. The idea is to get accurate gradient, this can be done with checkpoint-based methods (with a large integration error tolerance), or adjoint (but needs a small error tolerance).

---

> > > > > ### Author Response · Authors · 2021-08-17
> > > > > **Response**
> > > > >
> > > > > Thank you for further response.
> > > > >
> > > > > > Checkpointing is memory-consuming. Long-time integration is often necessary in practice, and you can't call it "non-practical".
> > > > >
> > > > > Sorry for confusion, but we referred to the situation "$f$ is small (e.g. 1-layer) but $z$ is large (e.g. ImageNet image size is 256x256x3)" as "non-practical"
> > > > >
> > > > > We agree that a long-time integration is necessary in practice *for prediction*. However, to our best knowledge, the number of steps *for training* is less than hundreds steps in most cases (FFJORD, physics simulations, image recognition, and so on). The checkpointing scheme requires memory only for training, and we confirmed that the memory for hundreds checkpoints does not dominate (see the previous response). Learning a time-series that needs thousands steps even using a high-order solver is an open problem.
> > > > >
> > > > > > High accuracy in integration is different from high-accuracy of model performance
> > > > >
> > > > > Sorry, you are right. MALI used a large tolerance value for training while obtaining the exact gradient. However, the symplectic adjoint method can do the same by using a second-order solver. With a large tolerance, the memory for checkpoint is almost negligible, so both are expected to work similarly. With a small tolerance, there is a trade-off; the symplectic adjoint method needs additional memory for checkpoints, but greatly reduces the computation time using a higher-order solver.
> > > > >
> > > > > We would like to emphasize that each of adjoint method, symplectic adjoint method, MALI, and ACA has its own advantages/disadvantages. These methods are complementary to each other; perfection in all terms is not required for acceptance. With a limited memory, the symplectic adjoint method is the best choice for continuous-time dynamics analysis (e.g., physics), MALI is the best for image and speech recognition, and both can work similarly for FFJORD. We will add this discussion in the final version.
> > > > >
> > > > > From this aspect, we believe that the symplectic adjoint method deserves a valuation at a similar level to the other methods.

---

> > ### Comment · Reviewer_G97n · 2021-08-17
> > **Can you run FFJORD on Imagenet on a single GPU?**
> >
> > Hi, thanks for your response.
> >
> > It occurs to me that a strong test for our discussion on memory requirement, is to run FFJORD on ImageNet. It has a pretty large $dim(z)$, also the model is pretty deep if I remember correctly, while all ffjord experiments in paper have a much lower dimension. My experience is, naive and ACA would cause memory explosion, only adjoint and MALI can run it (without modifying the default batchsize) on a single small-memory GPU (<12G). I wonder if the proposed method can work in large-scale problem. If possible, please report the batchsize and memory.

---

> > > ### Author Response · Authors · 2021-08-18
> > > **Theoretically Assumed**
> > >
> > > Thank you for keeping joining the discussion.
> > >
> > > > My experience is, naive and ACA would cause memory explosion, only adjoint and MALI can run it  (without modifying the default batchsize) on a single small-memory GPU (<12G).
> > >
> > > The original study of FFJORD did not examine ImageNet. When using the same experimental setting as MNIST and CIFAR-10, we confirmed that FFJORD for ImageNet *using the adjoint method* ran out the memory of 48 GB. As shown in my previous response, the adjoint method consumes the memory of around 1 GB for MNIST (28x28x1). For ImageNet (256x256x3), the same experimental setting is expected to consume the memory of 84 GB because the area is 84x larger. If you can run FFJORD for ImageNet using the adjoint method, some settings must be different.
> > >
> > > The memory consumption can be estimated theoretically. For ImageNet, the memory for the backpropagation is around 84x larger, and the memory for checkpoints is x251 times larger because of the difference in the number of channels. We suppose the number of steps is 1,000. Then, from my previous response, FFJORD is expected to consume the memory of 84 GB using the adjoint method and 118GB using the symplectic adjoint method. The memory consumption is less than double.
> > >
> > > Of course, one can suppose a situation where this difference is critical. We do not claim that the proposed method is universally superior. However, we believe that an issue under specific conditions does not diminish the value of the proposed method.
> > >
> > > The main drawback of ACA is not the memory for checkpoints but that for backpropagation. ACA performs the backpropagation through all stages of Runge-Kutta methods, while the symplectic adjoint method does it for a single use of a neural network. Hence, using DOPRI5, the symplectic adjoint method consumes 6 times less memory for the backpropagation than ACA. This is the main advantage of the symplectic adjoint method.

---

> > > > ### Comment · Reviewer_G97n · 2021-08-18
> > > > **Please try ImageNet 64x64**
> > > >
> > > > Sorry I did not specify the detail, please try ImageNet64 as https://github.com/cfinlay/ffjord-rnode or https://github.com/juntang-zhuang/TorchDiffEqPack, and report the batchsize, memory usage and rtol atol. I'm just curious what's the limit in practice. Would be better if you finish the training and report the final results.
> > > >
> > > > The drawback of both ACA and the proposed method is checkpoint memory compared to adjoint; symplectic-adjoint reduces memory compared to ACA but still grows with number of checkpoints. Whether the checkpoint memory is crucial varies from problem to problem.
> > > >
> > > > Please post your messages here whenever you have updates. Sorry I need to end here for the review, but in summary I think it has sufficient novelty, and my follow up responses are more like discussion to determine the limit in some really large-scale experiments, rather than criticism on the novelty.

---

> > > > > ### Author Response · Authors · 2021-08-18
> > > > > **Worked.**
> > > > >
> > > > > Thank you for the detail. Unfortunately, ImageNet and CelebA-HQ datasets used in the RNODE codes were removed from the websites, so we cannot reproduce them directly. Instead, we generated random images of 64x64x3, and we **set the number of checkpoints to 10,000**. It is enough to calculate the memory consumption. Even then, the RNODE codes with the symplectic adjoint method **successfully worked on GPUs with 12 GB memory**. Note that RNODE used 4 GPUs for this image size.
> > > > >
> > > > > We would like to respeak that, as the image area increases, the memory for checkpoints and the memory for the backpropagation increase at the same rate. Hence, **for images of any sizes,** the memory for hundreds of checkpoints is negligible, and the memory for thousands of checkpoints does not cause an issue.
> > > > >
> > > > > >  my follow up responses are more like discussion to determine the limit in some really large-scale experiments, rather than criticism on the novelty.
> > > > >
> > > > > We provided additional results and demonstrated that the memory for checkpoints does not cause an issue for "really large-scale experiments". Now, we believe that all your concerns about "the limit in some really large-scale experiments" have been addressed.
> > > > >
> > > > > > The drawback of both ACA and the proposed method is checkpoint memory compared to adjoint
> > > > >
> > > > > In addition, the additional drawback of ACA is backpropagation memory compared to the proposed method. And, this is why ACA caused memory explosion, while the proposed method did not.

---

> > > > > > ### Comment · Reviewer_G97n · 2021-08-18
> > > > > > **Thanks for your response**
> > > > > >
> > > > > > Thanks for addressing my concerns. I keep my rating for acceptance.

---

### Decision · Program_Chairs · 2021-09-27

**Decision:**

Accept (Poster)

**Comment:**

The paper proposes the use of the symplectic adjoint method for improving the memory footprint of neural ODE models. Reviewers agree that the use of such technique in the deep learning community is novel and supported by comprehensive numerical results.
I also believe that the memory consumption issue raised by reviewer rdkK has been, at least, partially addressed in the rebuttal.
I am therefore inclined to accept this work.